# Reciprocal impacts of telomerase activity and ADRN/MES differentiation state in neuroblastoma tumor biology

Eun Young Yu[1], Syed S. Zahid[1], Sarah Aloe[1], Erik Falck-Pedersen[1], Xi Kathy Zhou[2], Nai-Kong V. Cheung[3] &
Neal F. Lue [1,4] ✉

Telomere maintenance and tumor cell differentiation have been separately implicated in neuroblastoma malignancy. Their mechanistic connection is unclear. We analyzed neuroblastoma cell lines and morphologic subclones representing the adrenergic (ADRN) and mesenchymal (MES) differentiation states and uncovered sharp differences in their telomere protein and telomerase activity levels. Pharmacologic conversion of ADRN into MES cells elicited consistent and robust changes in the expression of telomere-related proteins. Conversely, stringent down-regulation of telomerase activity triggers the differentiation of ADRN into MES cells, which was reversible upon telomerase up-regulation. Interestingly, the MES differentiation state is associated with elevated levels of innate immunity factors, including key components of the DNA-sensing pathway. Accordingly, MES but not ADRN cells can mount a robust response to viral infections in vitro. A gene expression signature based on telomere and cell lineage-related factors can cluster neuroblastoma tumor samples into predominantly ADRN or MES-like groups, with distinct clinical outcomes. Our findings establish a strong mechanistic connection between telomere and differentiation and suggest that manipulating telomeres may suppress malignancy not only by limiting the tumor growth potential but also by inducing tumor cell differentiation and altering its immunogenicity.

[1] Department of Microbiology & Immunology, W. R. Hearst Microbiology Research Center, Weill Cornell Medicine, New York, NY, USA. [2] Department of Population Health Sciences, Weill Cornell Medicine, New York, NY, USA. [3] Department of Pediatrics, Memorial Sloan Kettering Cancer Center, New York, NY, USA. [4] Sandra and Edward Meyer Cancer Center, Weill Cornell Medical College, New York, NY, USA. ✉email: nflue@med.cornell.edu

Neuroblastoma (NB), a complex childhood cancer, has a unique biology of spontaneous remission and lethal progression depending on its genetic makeup[1,2]. Arising from pluripotent neural crest precursors of the sympathoadrenal lineage[3,4], NB has been known to manifest phenotypic diversity that mirrors the developmental potential of neural crest progenitor cells. In early studies of cell lines with distinct morphologies and biochemical markers, tumor cells were classified as N (neuroblastic), I (intermediate), and S (substrate-adherent) cells, and these cell types were shown to be interconvertible via pharmacologic treatment[5–8]. The different cell types were also found to exhibit varying degrees of tumorigenicity in animal models; the I cells, which manifest more features of stem cells, were particularly malignant[6]. More recently, detailed transcription and epigenetic profiling has reclassified NB tumors into the adrenergic (ADRN) and mesenchymal (MES) lineages that are defined by the activation of distinct super enhancers[9–11]. Inspection of the phenotypic features associated with the earlier morphologic/biochemical and the more recent genetic classifications suggests broad congruence between the two schemes, with the N and I cells showing substantial similarities to the ADRN lineage and the S cells resembling MES. For example, both S and MES cells harbor high levels of cytoskeletal proteins associated with the mesenchymal phenotype. Indeed, one pair of cell lines is shared between the earlier and more recent studies and their designations support the N/ADRN and S/MES groupings[5,10]. (For consistency, the ADRN and MES designations are used throughout this study for cell lines that were initially classified according to the N/I/S classification scheme.) Notably, multiple studies implicate the state of tumor cell differentiation in disease progression and treatment response. For example, the MES cells are more chemo-resistant and may be enriched in relapse and in metastatic diseases[10,12,13]. In contrast, the ADRN cells tend to predominate at diagnosis and show greater sensitivity to chemotherapies[10,11].

Another facet of NB biology that has been strongly implicated in disease progression is telomere maintenance[14,15]. Telomeres are dynamic nucleoprotein structures that stabilize the tips of chromosomes against aberrant DNA damage response and DNA repair[16]. As such, telomeres are crucial for both normal and cancer cell proliferation[17]. However, owing to incomplete end replication, telomere DNAs (a repetitive 6-base sequence) are subject to progressive loss with successive cell division and must be replenished to maintain an adequate reserve for cell proliferation. Indeed, recent studies of extensive tumor collections have confirmed a strong association between the presence of telomere maintenance mechanisms (TMMs) and high-risk disease as well as poor prognosis for NB[18,19]. The predominant TMM in NB is a special reverse transcriptase named telomerase, which uses the catalytic TERT protein in conjunction with an RNA template to add telomere repeats[20,21]. The second major TMM, known as ALT (alternative lengthening of telomeres), is a recombination pathway that resembles break-induced replication[22,23]. Consistent with the critical importance of TMMs, three of the most frequent genetic alterations in high-risk NB (MYCN amplification, TERT promoter rearrangement, and ATRX inactivation) are mutually exclusive and are each strongly linked to TMM activation—specifically, MYCN and TERT mutations give rise to high telomerase expression and ATRX loss triggers the ALT pathway[14,15,24]. The special importance of TMMs in NB may be a consequence of "telomere trimming"[25,26], a telomere shortening pathway that we recently demonstrated in a substantial fraction of high-risk NB tumors[27]. The need to compensate for the telomere loss precipitated by both incomplete end replication and telomere trimming may render NB especially dependent on TMMs. Besides telomerase and ALT-related

factors, many other proteins are known to regulate telomere length homeostasis and telomere protection. For example, the six-protein assembly (RAP1-TRF2-TIN2-TRF1-TPP1-POT1) that coats telomeres, known as shelterin, is crucial for suppressing aberrant DNA repair and DNA damage response at chromosome ends[16,28]. Although their roles in NB have not been investigated, dys-regulation of the shelterin proteins will probably impact tumor growth in line with observations for other cancers. There is thus growing interest in developing telomere and TERT-directed therapies against NB[29,30].

Remarkably, multiple studies have implicated telomere-related factors in neural development and neural differentiation. For example, the shelterin component TRF2 has been reported to regulate neural cell differentiation and neural protection, and these functions are at least partly mediated through nontelomeric pathways[31–33]. More specifically, a short, cytoplasmic isoform of TRF2 (TRF2-S) is thought to antagonize the silencing effect of REST to promote neural differentiation[31]. Thus, telomere proteins have the potential to alter the differentiation of neural crest-derived NB tumors in accordance with the developmental origin of these cells[3,4]. Likewise, telomerase and its catalytic protein subunit TERT may have similar potentials in NB differentiation given their proposed functions in neural development[34,35]. However, little is known about the roles of these telomere maintenance and protection factors in the ADRN and MES lineages of NB tumor cells.

In this study, we set out to address the roles of telomere-related factors in the different lineages of NB tumor cells. Using isogenic cells lines that display either ADRN or MES phenotypes, we showed that the levels of telomere factors vary dramatically, with the ADRN cells generally harboring much higher levels of such factors. Pharmacologically converting ADRN cells into MES cells triggered the expected changes in telomere protein profiles. Conversely, inhibiting telomerase activity in ADRN cells induced their conversion into MES cells in a reversible manner. Together, these findings establish a tight mechanistic connection between telomere regulation and NB lineage conversion. Another notable finding was the strong upregulation of DNA-sensing and innate immunity factors in MES cells, which supports several recent reports of similar associations[36,37]. By linking telomere regulation to tumor cell differentiation and immunogenicity in NB, our study provides potential explanations for the disparate clinical outcomes of NB cases, and reinforces the rationale for developing telomere-directed therapies against this aggressive pediatric cancer.

## Results

**ADRN and MES cells are distinguished by prominent differences in telomere proteins.** Three matched pairs of ADRN and MES cell lines (LA1-55N [ADRN] and LA1-5S [MES]; LA1-66N [ADRN] and LA1-6S [MES]; SH-SY5Y [ADRN] and SH-EP1 [MES]) were analyzed with respect to the levels of previously defined lineage markers (e.g., SLUG and HES1 for MES and GATA3 for ADRN) as well as telomere-related proteins (Fig. 1a, b). As expected, the different cell lines preferentially express the associated differentiation markers (Fig. 1a). Also similar to previous reports, the levels of MYCN are high in ADRN cells harboring MYCN amplification[6]. Notably, multiple telomere-related factors were found to be differentially expressed in ADRN and MES cells (Fig. 1b). These include three shelterin components that were over-expressed in ADRN cells relative to MES (TRF1, TRF2 and TPP1), and another component that was elevated in MES (POT1). Also differentially expressed are other factors implicated in telomere or chromosome maintenance. In general, proteins involved telomere DNA synthesis or repair (ATRX,

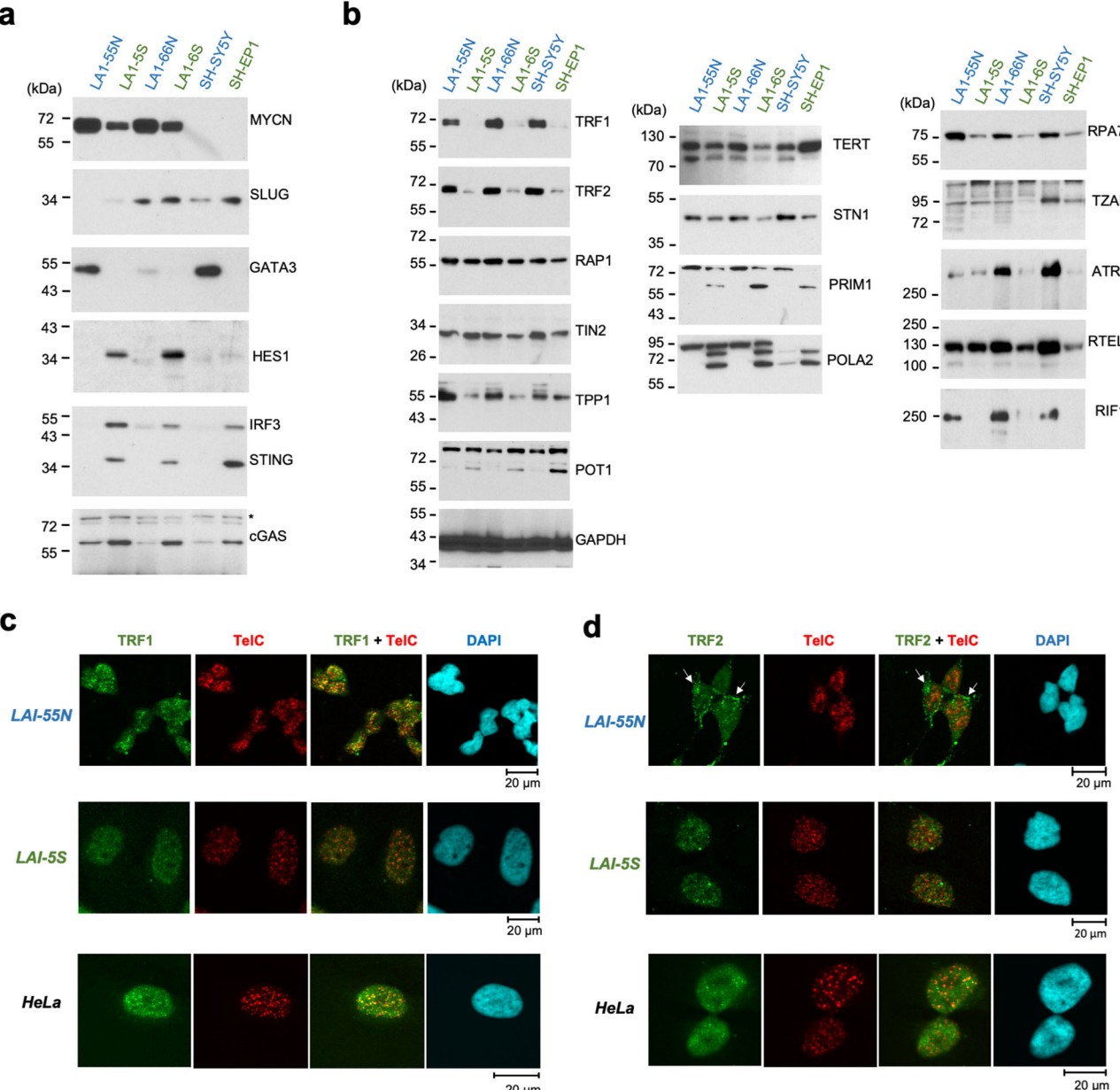

**Fig. 1 Matched ADRN and MES cell lines show prominent differences in telomere protein expression and localization. a** and **b** Western analysis of cell lineage, DNA sensing, and telomere-related factors in three matched ADRN and MES cell lines. The LA1-55N (ADRN) and LA1-5S (MES) pair, as well as the LA1-66N (ADRN) and LA1-6S (MES) pair, were both derived from LAN-1, a MYCN-amplified NB tumor. The SH-SY5Y (ADRN) and SH-EP1 (MES) pair was derived from a non-MYCN-amplified tumor. ADRN and MES cell line names are displayed in blue and green, respectively, in all figures in this study. **c** and **d** IF-FISH analysis of TRF1 and TRF2 in the indicated cell lines. Prominent examples of peri-nuclear staining of TRF2 in LA1-55N are marked with white arrows.

RIF1, RPA70, RTEL1, STN1) tend to be more abundant in ADRN cells. However, two subunits of the primase-Pol α complex (PRIM1 and POLA2), which are important for telomere replication as well as telomere C-strand extension[38,39], are more abundant in MES cells. These results suggest that the telomere nucleoprotein structure may undergo remodeling during tumor cell differentiation between ADRN and MES cells.

To determine if changes in telomere protein levels are associated with changes in telomere nucleoprotein configuration, we analyzed the cellular distribution of TRF1 and TRF2, the two major double-strand telomere binding proteins in the shelterin complex (Fig. 1c, d, Supplementary Fig. 1a). Consistent with their overall protein levels in Western analysis, both TRF1 and TRF2

displayed stronger signals and brighter foci in the ADRN cell line LA1-55N. As expected, TRF1 was exclusively in the nucleus and exhibited substantial co-localization with telomere foci, especially in LA1-55N. In contrast, TRF2 displayed prominent nonnuclear staining in a cell type-specific manner. In LA1-55N, TRF2 was present in both the nucleus and the cytoplasm, and showed especially bright cytoplasmic foci that cluster in the peri-nuclear region (Fig. 1d and Supplementary Fig. 1a). Moreover, the nuclear signals of TRF2 were predominantly non-telomeric, suggesting that the telomere chromatin of ADRN cells contain low levels of TRF2 despite the overall abundance of this protein (see also results below for another ADRN cell line BE(2)N (abbreviated from SK-N-BE(2)N)). In LA1-5S, TRF2 showed

faint nuclear staining that overlapped slightly with telomere foci. Parallel staining of TRF1 and TRF2 in HeLa cells revealed the expected nuclear localization and signal overlap with telomere foci (Fig. 1c, d), indicating that the unusual TRF2 pattern in ADRN cells is unlikely to be an artifact of staining. One possible explanation for the cytoplasmic TRF2 signal comes from earlier reports of TRF2-S, a short isoform found in the cytoplasm of neural cells[31]. In support of the presence of TRF2-S in NB, longer exposures of TRF2 Western blots revealed the presence of a 34 kDa immune-reactive protein that is reminiscent of TRF2-S (Supplementary Fig. 1b). Overall, the immunostaining supports the notion that there are lineage-specific differences between the telomere nucleoprotein structures of ADRN and MES cells.

The DNA-sensing pathway has been linked to telomere DNA damage and the ALT mechanism of telomere maintenance[40,41]. We analyzed several key components of this pathway in our cell lines, and observed dramatic upregulation of cGAS, STING and IRF3 in MES cells (Fig. 1a), suggesting that these cells may be more responsive to abnormal DNA stimuli. Other groups have recently reported differences in the immune profiles of ADRN and MES cells[36,37].

**Telomere lengths and telomerase activity differ between ADRN and MES cells, but exhibit no uniform pattern with respect to cell lineage**. To determine if the differences in telomere proteins in ADRN and MES are associated with changes in telomere maintenance, we first analyzed telomere lengths in the respective cell lines using both standard Southern analysis and STELA, a PCR-based assay that preferentially detects short telomeres (Fig. 2a–c)[27]. While each pair of cell lines showed some differences in the length distribution of telomeres, there was no consistent pattern with respect to cell lineage, even for matched ADRN and MES lines derived from the same tumor. Specifically, whereas LA1-55N (ADRN) harbored shorter telomeres than the matched LA1-5S (MES), the opposite was true for LA1-66N (ADRN) and LA1-6S (MES). We also analyzed single stranded DNA at telomeres, which reflect different aspects of telomere regulation; elevated C-strand is often due to "telomere trimming"[25,26,42] and elevated G-strand is a marker of telomere de-protection[43]. Some differences between ADRN and MES were observed for C-strand ssDNA, but these differences were again not consistently related to the cell lineage (Fig. 2d). Finally, telomerase activities in the cell lines were assessed and were found to manifest MYCN-dependent cell lineage differences. In particular, telomerase activity was down regulated by 10–15 fold in matched MES cells, but only in *MYCN*-amplified tumor cell lines, i.e., in LA1-5S and LA1-6S but not in SH-EP1 (Fig. 2e). While tumor cell lineage does not strictly control telomere lengths or telomerase activity, the combined effects of telomere lengthening and shortening pathways on overall telomere lengths may impact cell growth. Among the lines derived from LAN-1, LA1-6S has the lowest telomerase activity and highest level of telomere trimming, which may explain its having the shortest telomeres and the slowest growth rate (Fig. 2c and Supplementary Fig. 1c).

**Conversion of ADRN into MES cells is accompanied by telomere remodeling**. Chronic low dose BrdU treatment has been shown to effectively trigger the conversion of ADRN cells into MES cells[7,8]. Indeed, when cultured in BrdU, the ADRN cell line BE(2)N manifested progressive morphologic changes that culminated in an MES-like state by day 26 (Fig. 3a). Notably, this conversion is probably not just due to the selection of a preexisting pool of MES cells in the BE(2)N cell population; proliferation analysis indicates that there was little increase in cell number beyond eight days of BrdU treatment, whereas the morphologic changes continued from

day 8 through day 20 (Supplementary Fig. 2a, b). In addition, the level of the MES marker STING increased by ~3 fold from D14 to D20, while there was no increase in cell number or detectable cell death (Supplementary Fig. 2c). While we cannot rule out some degree of selection during the early phase of BrdU treatment, our results indicate that the BrdU effects on differentiation cannot be explained by selection alone. Consistent with gradual conversion to the MES lineage, telomerase activity in the cell population declined early and progressively to become nearly undetectable (>100-fold reduced at day 26) (Fig. 3b). The levels of telomere factors, STING and MYCN likewise manifested the expected changes toward the MES expression profile (Fig. 3c). Localization of TRF1 and TRF2 at early and late time points of BrdU treatment mirrored their distributions in ADRN and MES cells, respectively, with TRF2 showing peri-nuclear/cytoplasmic staining on day 4 and faint nuclear staining on day 16 (Supplementary Fig. 3a, b). Interestingly, the decline in TRF1, TRF2 and MYCN levels occurred rapidly within the first five days, whereas the increase in STING was especially prominent during the late stage of conversion (after day 10 and continues beyond day 20). These findings corroborate the differences in telomere proteins and telomerase activity observed in naturally derived ADRN and MES cell lines, and support a progressive change in the gene expression program during BrdU-induced phenotypic conversion. Interestingly, the reduction in telomerase activity is not associated with detectable changes in telomere lengths (Fig. 3d), echoing the inconsistent relationship between cell lineage and telomere lengths in naturally derived ADRN and MES cell lines.

To more comprehensively characterize gene expression changes during lineage conversion, we used RNA-seq to profile the transcriptomes of BE(2)N at multiple time points during BrdU treatment. In support of the notion of gradual phenotypic conversion, we found that a progressively higher number of genes were upregulated from early to late time points, and this applies to downregulated genes as well (Fig. 3e). The heatmaps of individual genes further support progressive changes in the level of mRNAs (Fig. 3f). Comparison of the RNA-seq data to previously characterized signatures revealed substantial overlap between genes upregulated by BrdU and the MES signature, as well as an overlap between genes downregulated by BrdU and the ADRN signature (Fig. 3g). Moreover, pathway analysis of genes upregulated in MES cells highlighted the involvement of cell adhesion/motility, epithelial-mesenchymal transition, and immune regulatory genes in the phenotypic switch (Fig. 3h), further reinforcing the resemblance between naturally derived and BrdU-induced MES cells.

We tested the generality of our observations on BE(2)N by analyzing two other MYCN-amplified, ADRN-like cell lines: BE(2)C (similar to BE(2)N and thought to exhibit more stem cell characteristics) and SK-N-HM (derived from the brain metastasis of another *MYCN*-amplified NB case)[7]. Just like BE(2)N, BrdU-treated BE(2)C displayed prominent changes in telomere, DNA-sensing, and lineage-related proteins (e.g., decreases in TRF1, ATRX, MYCN, and GATA3, as well as increases in PRIM1, POLA2, IRF3, STING, and SLUG) (Supplementary Fig. 4a). Also resembling BE(2)N, while telomerase activity was greatly reduced by BrdU in BE(2)C, no change in telomere length was observed (Supplementary Fig. 4b, c). We considered the possibility that this maintenance of telomere length (despite telomerase activity reduction) could be due to ALT activation. However, the level of C-circles, a marker of ALT activity, was unchanged in BrdU-treated BE(2)C (Supplementary Fig. 4d, left panel). In addition, similar levels of C-circles were detected in matched ADRN and MES cell lines, indicating that MES cells do not utilize ALT for telomere maintenance (Supplementary Fig. 4d, right panel). SK-N-HM also responded to BrdU by switching to an MES-like

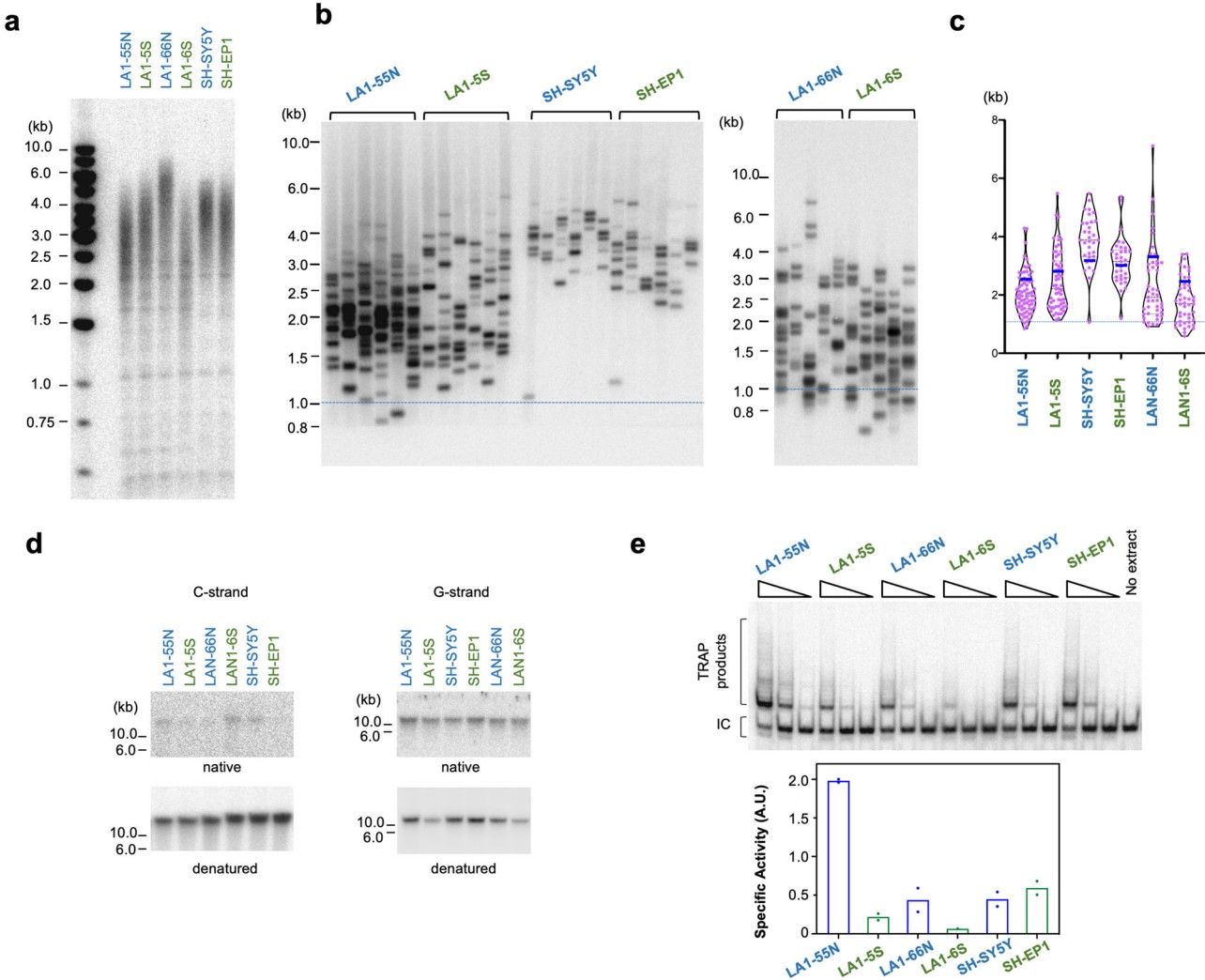

**Fig. 2 Characterization of telomere lengths, telomere ssDNA, and telomerase activity in matched ADRN and MES cell lines. a** The telomeres of the indicated cell lines were subjected to TRF Southern analysis. **b** The telomeres of the indicated cell lines were subjected to STELA analysis. Extra-short telomeres (<1 Kb) are demarcated by horizontal red lines across the panels. **c** The STELA fragments from each cell line in **b** (n = 6 PCR reactions) were analyzed using TeSLA software and the pooled results presented as violin plots with medians and quartiles. **d** The amount of ssDNA in the telomere G- and C-strand in the indicated cell lines were analyzed by native in-gel hybridization. **e** Telomerase activities in the indicated cell lines were analyzed by TRAP and plotted (means, n = 2 independent extracts).

morphology and exhibiting a reduction in telomerase activity. However, the change in telomere and DNA sensing protein levels in this cell line was slower and milder than those in BE(2)N and BE(2)C (Supplementary Fig. 4e, f). Overall, our data indicate that the telomere-related differences observed between naturally derived ADRN and MES cell lines are recapitulated in pharmacologically generated ADRN and MES pairs.

**Stringent inhibition of telomerase activity triggers the reversible conversion of ADRN into MES cells.** The BrdU experiment showed that telomerase activity declined early and progressively during the conversion of ADRN cells into MES cells, suggesting that this activity may be involved in NB tumor cell differentiation, especially in *MYCN*-amplified tumor cells. To test this idea, we inhibited telomerase in the BE(2)N cells by expressing a catalytically inactive, dominant negative allele of hTERT (Dn-hTERT)[44]. We generated amphotropic retroviruses carrying Dn-hTERT or wild-type hTERT, and infected BE(2)N cells with these viruses. Following drug selection, the cell populations were collected and passaged until ~day 100. Telomerase upregulation and

inhibition (in cell populations harboring hTERT and Dn-hTERT, respectively) were confirmed by the TRAP activity assays (Fig. 4a). In multiple, independently propagated Dn-hTERT-harboring cell populations, telomerase activity was greatly reduced, to ~5% of the level found in parental BE(2)N (day 55 samples in Fig. 4a and Supplementary Fig. 5a). However, with further passaging, the activity became derepressed (to ~50% of the parental level or more). This derepression may be due to a reduction in the level of Dn–hTERT mRNA, which was ~8-fold lower on day 82 in comparison to day 55 as judged by allele–specific RT–PCR (Supp Fig. 5b). While the underlying reason for the reduction in Dn–hTERT mRNA is unclear, the resulting derepression of telomerase activity enabled us to examine the reversibility of the phenotypes induced by telomerase inhibition (see below). As expected, cells infected with the hTERT-bearing viruses manifested high levels of telomerase activity throughout passages (Fig. 4a).

We first characterized telomere lengths in the infected cells during passage. As anticipated, the increase in telomerase activity in hTERT-expressing cells caused telomere elongation, whereas the decrease in telomerase activity in Dn-hTERT-expressing cells

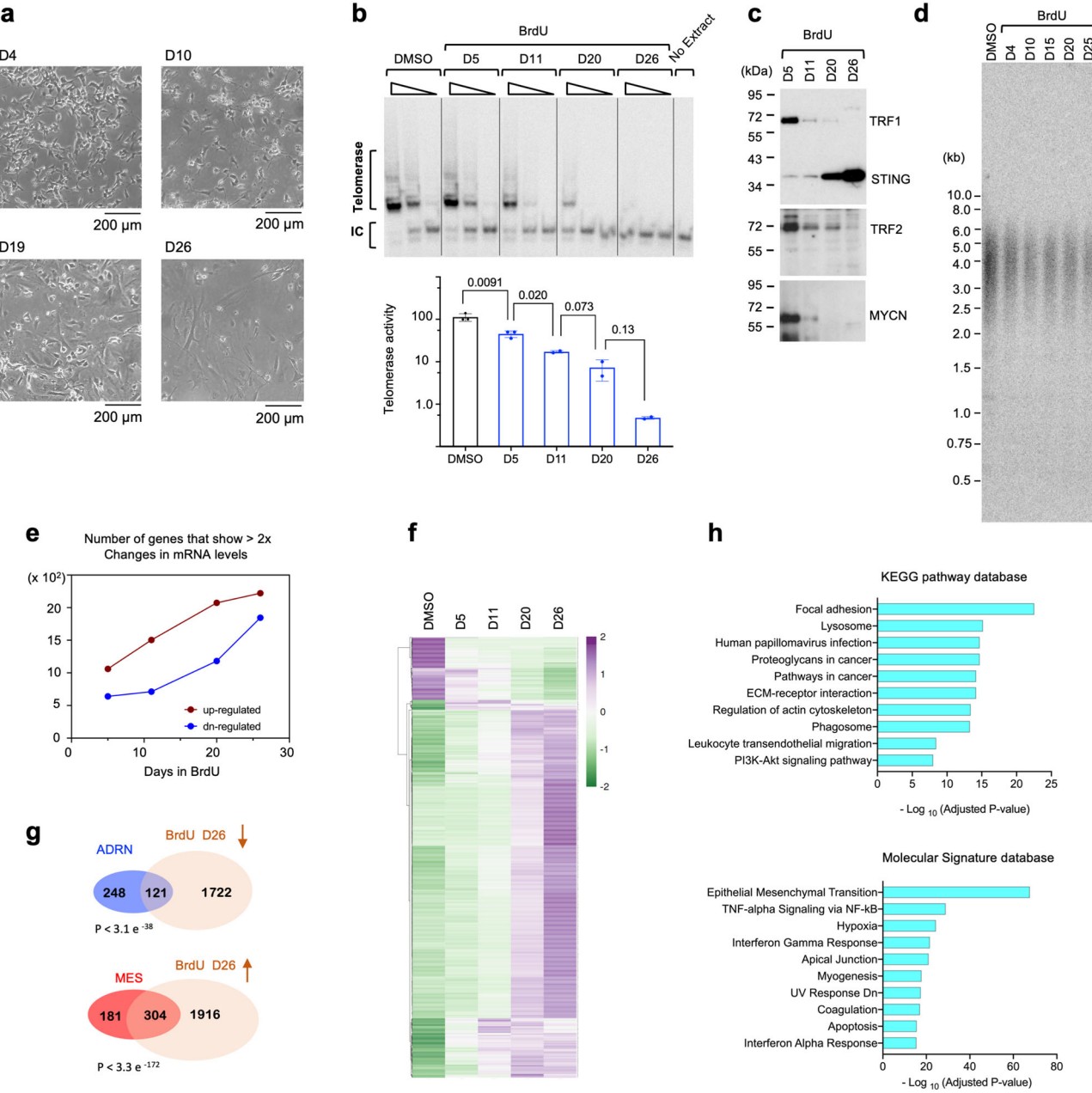

**Fig. 3 Profiling of BE(2)N cells during BrdU-induced switch from ADRN to MES cell types. a** Morphology of BE(2)N at different time points following BrdU treatment. (**b**) TRAP analysis of telomerase activity in BE(2)N at different time points during BrdU-induced phenotypic switch. The assays were performed using serial dilutions of the extracts (equivalent to 500, 50 and 5 ng of proteins), and the relative activities (mean ± S.D., $n = 2$ or 3 experimental replicates) are plotted at the bottom. Pair-wise P-values were determined using student's t-tests (two-tailed). **c** Western analysis of telomere and lineage-related proteins in BE(2)N during BrdU treatment. **d** Analysis of telomere length distributions in BE(2)N at different time points during BrdU-induced phenotypic switch. **e** The number of upregulated and downregulated genes in BrdU-treated BE(2)N (by >2-fold in comparison to the DMSO-treated control cells) were determined from RNA-seq analysis and plotted. **f** The expression patterns of the ~1100 genes that show the greatest changes in mRNA levels (>3 fold) in BrdU-treated BE(2)N were displayed using Heatmap. **g** The list of genes that were downregulated in BrdU-treated BE(2)N (by >2-fold) was compared to the ADRN signature list[10]. In parallel, the list of genes that were upregulated in BrdU-treated BE(2)N (by >2-fold) was compared to the MES signature. **h** The ~2000 genes upregulated in BrdU-treated BE(2)N (by >2-fold) were subjected to pathway analysis by the Enrichr program. The top ten GO terms identified by the KEGG pathway database and the Hallmark gene sets in the Molecular Signature Database were ranked by adjusted P value and plotted.

caused telomere shortening (Fig. 4b). Surprisingly, the derepression of telomerase in late passages of Dn-hTERT-containing cells did not restore telomere length. Most interestingly, we found that telomerase inhibition can trigger the morphologic conversion of BE(2)N from ADRN into MES-like cells (Supplementary Fig. 5c). Similar to BrdU-induced conversion, the MES-like cells elicited by Dn-hTERT displayed the MES protein expression profile (Fig. 4c); the levels of telomere factors (TRF1 and TRF2) and MYCN were substantially reduced and that of STING was greatly increased at day 55 and 64. Remarkably, the time-dependent reversal in telomerase inhibition is matched by a similar reversion in cell morphology and protein expression profiles. In particular,

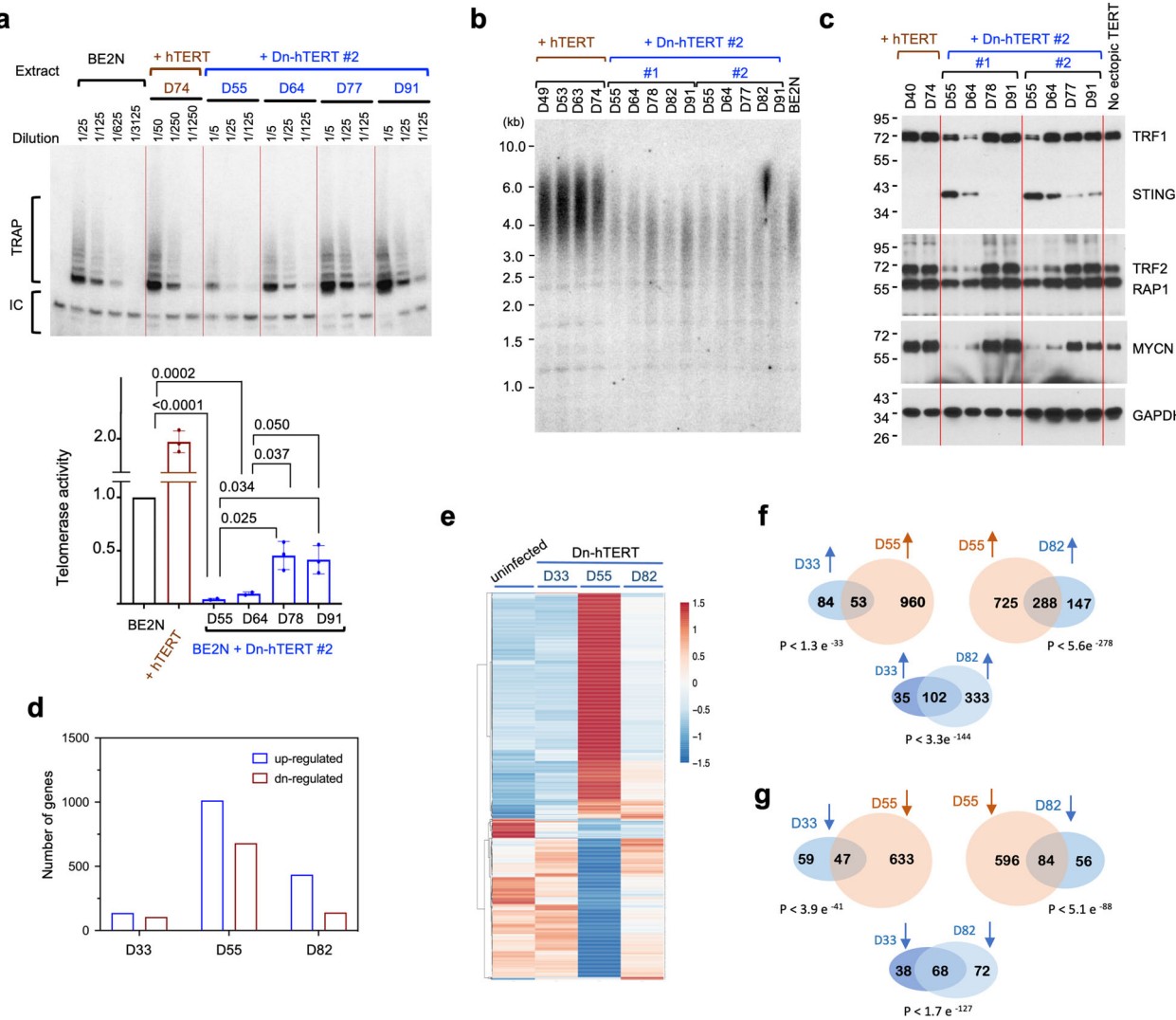

**Fig. 4 Profiling of BE(2)N cells during Dn-hTERT-induced switch from ADRN to MES cell types. a** TRAP analysis of telomerase activity in BE(2)N harboring either hTERT or Dn-hTERT at different time points following retrovirus infection. The assays were performed using serial dilutions of the extracts as indicated. The relative activity of each extract was quantified using ImageQuant and plotted at the bottom (mean ± S.D., n = 2 or 3 experimental replicates). Pair-wise P-values were determined using student's t-tests (two-tailed). **b** Analysis of telomere length distributions in BE(2)N harboring either hTERT or Dn-hTERT at different time points following retrovirus infection. **c** Western analysis of telomere and lineage-related proteins in BE(2)N during passage of cells infected with viruses that harbor either hTERT or Dn-hTERT. **d** The number of upregulated and downregulated genes in Dn-hTERT-treated BE(2)N at the indicated time points (by >2-fold in comparison to the DMSO-treated control cells) were determined from RNA-seq analysis and plotted. **e** The expression patterns of the ~1000 genes that show the greatest changes in mRNA levels (>2 fold) in BrdU-treated BE(2)N were displayed using Heatmap. **f** Pair-wise comparison of the degrees of overlaps between genes upregulated by Dn–hTERT treatment at different time points. **g** Pair–wise comparison of the degrees of overlaps between genes downregulated by Dn-hTERT treatment at different time points.

low telomerase activity at day 55 and 64 correlated with MES cell morphology (enlarged and flattened with prominent vesicles) and protein profiles, whereas high telomerase activity on day 77 and 91 was associated with the opposite features (Fig. 4c, Supplementary Fig. 5c). Our observations strongly suggest that telomerase activity controls NB cell conversion in a reversible manner.

We also tested the effect of Dn-hTERT on BE(2)C, which resembles BE(2)N phenotypically but is thought to possess more stem cell characteristics. However, telomerase inhibition in BE(2) C cells (~70% inhibition) was not as efficient as that in BE(2)N, and telomerase activity was rapidly derepressed to >50% of the normal level found in parental cells (Supplementary Fig. 6). Notably, the Dn-hTERT-treated BE(2)C did not manifest morphologic conversion to MES-like cells, suggesting that

stringent inhibition of telomerase is required for NB lineage switch. It is worth noting that in BrdU-treated BE(2)N, clear MES-like morphology did not emerge until ~day 20, when telomerase activity was less than 10% of untreated cells, again suggesting that stringent inhibition of telomerase is needed for full phenotypic conversion.

To characterize the genome-wide gene expression pattern during Dn-hTERT-induced lineage conversion, we compared the transcriptomes of Dn-hTERT-expressing cells at multiple time points by RNA-seq (Fig. 4d–f). Similar to the transcriptomes of BrdU-treated BE(2)N, we found that telomerase inhibition induced progressive changes in gene expression patterns that correlated kinetically with changes in morphology and protein profiles. In particular, more genes were up or downregulated in Dn-hTERT-treated cells on day 55 (showing the greatest change

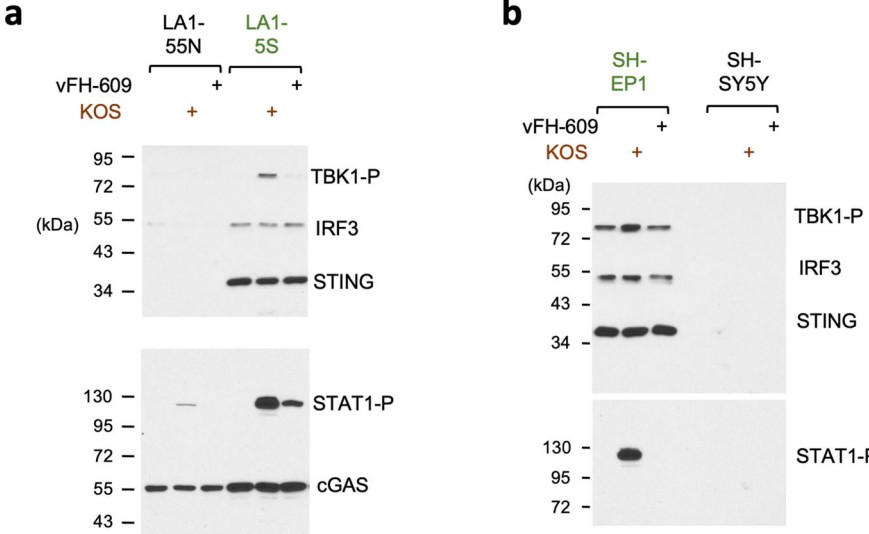

**Fig. 5 Activation of the DNA sensing pathway in ADRN and MES cells by Herpes virus. a** Western analysis of factors in the DNA sensing pathway in LA1-55N and LA1-5S cell lines following Herpes virus infection. **b** Western analysis of factors in the DNA sensing pathway in SH-SY5Y and SH-EP1 cell lines following Herpes virus infection.

in morphology) than on day 33 or day 82 (Fig. 4d). The heatmaps of individual genes further support progressive changes in the level of mRNAs from day 33 to day 55, as well as the reversal of expression on day 82 (Fig. 4e). Also consistent with progressive change, comparisons of the RNA-seq data at different time points showed substantial overlaps in the upregulated and down-regulated gene lists between each pair of adjacent time points (i.e., between day 33 and day 55, as well as between day 55 and day 82, Fig. 4f). Moreover, the strong overlaps between the up- and downregulated gene sets on day 33 and day 82 indicate that morphologic reversion is due to the reversal of the transcription programs in these cells. Interestingly, comparison of the protein and RNA profiles suggests that some of the differences between cell lineages may be due to post-transcriptional regulation. For example, while the MYCN and STING mRNA levels changed in parallel with the respective proteins, the TRF1 and TRF2 mRNAs were unchanged throughout the Dn-hTERT-induced conversion process, suggesting that these telomere proteins are regulated at a post-transcriptional step.

We next compared the upregulated and downregulated gene lists in the Dn-hTERT-induced MES-like cells (day 55) to the corresponding gene lists in BrdU-induced MES cells, and with previously defined MES and ADRN signature genes (Supple-mentary Fig. 7a)[10]. Strong overlaps were observed in each case, supporting the correspondence between BrdU- and Dn-hTERT-induced MES cells, as well as the correspondence between these cells and previously defined MES. Gene ontology analysis of factors upregulated by telomerase inhibition likewise support the involvement of MES-related pathways such as cell adhesion and immune response (Supplementary Fig. 7b). We also characterized the expression of a subset of ADRN and MES signature genes in our BrdU and Dn-hTERT experiments and found the ADRN genes to be consistently downregulated by these treatments and the MES genes to be consistently upregulated (Supplementary Fig. 7c). Altogether, these analyses further support the designa-tions of the different cell types in our study as ADRN and MES.

**MES but not ADRN cells harbor high levels of DNA sensing factors and activate the innate immunity pathway in response to DNA virus infection.** To address the functional impact of the

over-expression of DNA-sensing/innate immunity factors in MES cells, we infected two pairs of ADRN/MES cells with wild type Herpes Simplex Virus (KOS) or a mutant with altered cellular trafficking (vFH609)[45]. Indeed, both MES cell lines (LA1-5S and SH-EP1) exhibited robust response to KOS as evidenced by increased phosphorylation of TBK1 and STAT1, while the cor-responding ADRN cells (LA1-55N and SH-SY5Y) manifested little or no phosphorylation of these proteins (Fig. 5a, b). As expected from previous studies, vFH609 was less active than KOS in stimulating the phosphorylation cascade. Thus, the elevated levels of DNA-sensing and innate immunity factors in MES evidently allow these cells to mount a stronger immunogenic response to DNA stimuli.

**Evidence for NB tumor cells with ADRN and MES expression profiles and relationship to prognosis.** To assess the relevance of the ADRN and MES cell lineages to NB tumors in patients, we queried the expression profiles of NB tumors in the Pediatric Neuroblastoma Target 2018 collection, specifically a set of tumors with RNA-seq data ($n = 137$) at cBioportal (https://www.cbioportal.org/) (see Table 1 for patient characteristics). Given the strong relationships between telomeres, cell lineage, and innate immunity in NB, the expression patterns (rank transformed) of 29 genes composed of key components of these pathways were used to identify tumors with predominantly ADRN or MES characteristics (Fig. 6a, b). Three clusters were uncovered through PAM clustering analysis, with clusters 1 and 3 ($n = 35$ and $n = 50$) manifesting ADRN-like profiles (i.e., high telomere/low immunity/low MES marker expression) and cluster 2 ($n = 52$) manifesting an MES-like profile (i.e., low telomere/high immunity/high MES marker expression) (Table 1 and Fig. 6b). This finding supports the existence of NB tumors composed of predominantly one or the other cell types. We also analyzed the relative expression of specific telomere proteins against immunity and cell lineage markers in these samples and confirmed the expected positive and negative correlations (Sup-plementary Fig. 8). For example, TRF2 expression is negatively correlated with STING and NOTCH3, but positively correlated with PHOX2B, precisely as predicted given the expression of these genes in MES and ADRN cell lines. We then explored the

**Table 1 Patient characteristics in the three gene expression clusters.**

| | All (n = 137) | 1(n = 35) | 2(n = 52) | 3(n = 50) | p-values* |
|---|---|---|---|---|---|
| AGE | | | | | |
| Mean+/−sd | 3.99+/−2.89 | 2.91+/−2.29 | 4.1+/−3.42 | 4.62+/−2.46 | |
| Median (IQR) | 3 (2, 5) | 2 (1,4) | 3 (2,5) | 4 (3,6) | 0.001 |
| SEX, n (%) | | | | | |
| Female | 57 (41.61%) | 8 (22.86%) | 25 (48.08%) | 24 (48%) | |
| Male | 80 (58.39%) | 27 (77.14%) | 27 (51.92%) | 26 (52%) | 0.031 |
| RACE, n (%) | | | | | |
| Asian | 1 (0.73%) | 0 (0%) | 0 (0%) | 1 (2%) | |
| Black or African American | 24 (17.52%) | 8 (22.86%) | 7 (13.46%) | 9 (18%) | |
| Native Hawaiian or other Pacific Islander | 2 (1.46%) | 0 (0%) | 0 (0%) | 2 (4%) | |
| Not Reported | 3 (2.19%) | 1 (2.86%) | 1 (1.92%) | 1 (2%) | |
| Unknown | 10 (7.3%) | 4 (11.43%) | 2 (3.85%) | 4 (8%) | |
| White | 97 (70.8%) | 22 (62.86%) | 42 (80.77%) | 33 (66%) | 0.528 |
| ETHNICITY, n (%) | | | | | |
| Hispanic or Latino | 12 (8.76%) | 4 (11.43%) | 3 (5.77%) | 5 (10%) | |
| Not Hispanic or Latino | 114 (83.21%) | 27 (77.14%) | 44 (84.62%) | 43 (86%) | |
| Unknown | 11 (8.03%) | 4 (11.43%) | 5 (9.62%) | 2 (4%) | 0.554 |
| INSS_STAGE, n (%) | | | | | |
| Stage 2b | 1 (0.73%) | 1 (2.86%) | 0 (0%) | 0 (0%) | |
| Stage 3 | 5 (3.65%) | 3 (8.57%) | 1 (1.92%) | 1 (2%) | |
| Stage 4 | 111 (81.02%) | 20 (57.14%) | 43 (82.69%) | 48 (96%) | |
| Stage 4 s | 20 (14.6%) | 11 (31.43%) | 8 (15.38%) | 1 (2%) | <0.001 |
| TUMOR_SAMPLE_HISTOLOGY, n (%) | | | | | |
| Favorable | 25 (18.25%) | 12 (34.29%) | 10 (19.23%) | 3 (6%) | |
| Unfavorable | 102 (74.45%) | 21 (60%) | 36 (69.23%) | 45 (90%) | |
| Unknown | 10 (7.3%) | 2 (5.71%) | 6 (11.54%) | 2 (4%) | 0.005 |
| DIAGNOSIS, n (%) | | | | | |
| Ganglioneuroblastoma, intermixed | 1 (0.73%) | 0 (0%) | 1 (1.92%) | 0 (0%) | |
| Ganglioneuroblastoma, nodular | 22 (16.06%) | 5 (14.29%) | 10 (19.23%) | 7 (14%) | |
| Neuroblastoma | 109 (79.56%) | 27 (77.14%) | 40 (76.92%) | 42 (84%) | |
| Unknown | 5 (3.65%) | 3 (8.57%) | 1 (1.92%) | 1 (2%) | 0.567 |
| RISK_GROUP, n (%) | | | | | |
| High Risk | 113 (82.48%) | 20 (57.14%) | 44 (84.62%) | 49 (98%) | |
| Intermediate Risk | 10 (7.3%) | 8 (22.86%) | 1 (1.92%) | 1 (2%) | |
| Low Risk | 14 (10.22%) | 7 (20%) | 7 (13.46%) | 0 (0%) | <0.001 |
| OS_STATUS, n (%) | | | | | |
| 0:LIVING | 64 (46.72%) | 21 (60%) | 27 (51.92%) | 16 (32%) | |
| 1:DECEASED | 73 (53.28%) | 14 (40%) | 25 (48.08%) | 34 (68%) | 0.024 |
| OS_DAYS | | | | | |
| Mean+/−sd | 1622.13+/−1094.61 | 1826.94+/−1162.02 | 1727.88+/−1176.46 | 1368.78+/−916.92 | |
| Median (IQR) | 1544 (723, 2325) | 2064 (806,2500) | 1543.5 (805.75,2267.5) | 1323.5 (487.25,2012.75) | 0.106 |
| OS_MONTHS | | | | | |
| Mean+/−sd | 53.81+/−35.97 | 60.46+/−38.14 | 57.29+/−38.68 | 45.54+/−30.19 | |
| Median (IQR) | 51 (24, 77) | 68 (27,82.5) | 51 (26.75,75.25) | 44 (16.75,67) | 0.112 |

*p-values were based on the nonparametric Kruskal–Wallis test for continuous variables and Fisher's exact test for categorical variables.

potential impact of cell lineages in disease progression by comparing the outcomes of patients belonging to different clusters (Fig. 6c). In univariate analysis, the cluster 1 and cluster 3 patients, despite both having predominantly ADRN-like profiles, manifested the best and worst overall survival (OS), respectively ($p = 0.013$). The MES-like cluster 2 patients had an intermediate outcome, although the difference between cluster 2 and 3 ($p = 0.056$) did not meet the threshold for statistical significance. Notably, after adjusting for age and stage of disease (two previously established prognostic variables), the survival differences between the different clusters were no longer statistically significant. Together, these results suggest that NB cell lineage may be correlated with stage (regressing/differentiating local regional versus metastatic/relapsing) and age at diagnosis while influencing treatment outcome.

To identify differences between cluster 1 and 3 tumors that are associated with the disparity in outcome, we queried for genes that showed significant differences in RNA levels across the two clusters ($p < 0.01$, Wilcoxon rank sum test). Eight genes in our list of 29 were found to fit this criterion, and seven of these were related to telomere regulation (Supplementary Table 1). Among the hits was TERT, which had previously been established as a strong prognostic indicator[19]. We further explored the role of TERT expression by examining its predictive values in both univariate and multivariable analysis (adjusting for age and stage using Cox proportional hazards model), and observed better OS for patients with low levels of TERT (below median expression) in both analyses ($p < 0.001$ and $p = 0.021$) (Fig. 6d). Similar results were obtained for event-free survival (EFS) (Fig. 6d), except that the EFS difference in multivariable analysis was not statistically significant. Overall, these comparisons suggest that differences in TMM may be a key contributor to the differential outcomes of cluster 1 and 3 patients.

We also applied our gene signature analysis to a second group of patients (also in the Pediatric Neuroblastoma Target 2018 collection), whose tumors have been subjected to microarray

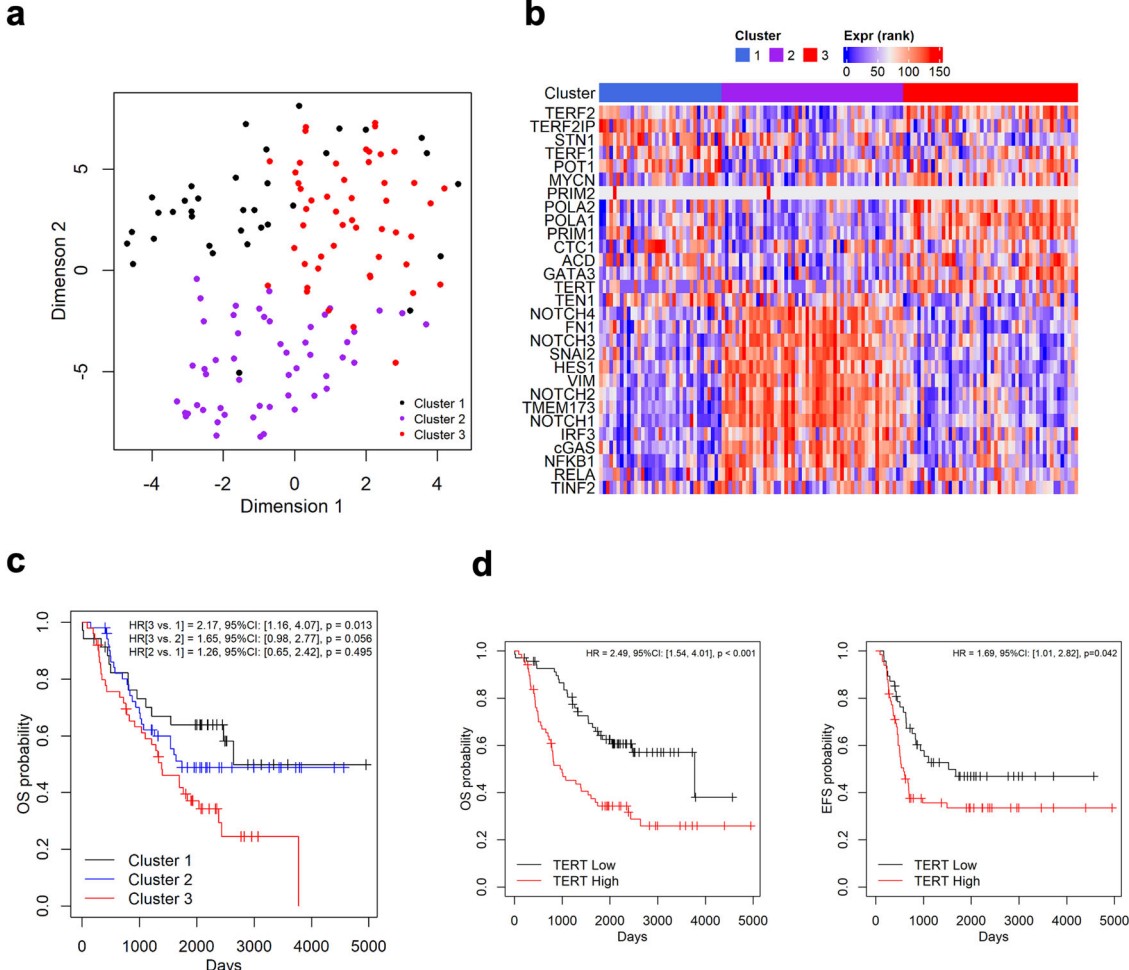

**Fig. 6 Analysis of clinical tumor samples using a telomere- and cell lineage-related signature gene list. a** The RNA levels (rank transformed) of a list of 29 telomere-, cell lineage- and immunity-related genes across 137 NB tumors with RNA-seq data were analyzed and clustered using PAM and displayed as a t-SNE plot. **b** The rank transformed RNA levels of signature genes across clustered NB tumor samples are displayed in Heatmap. **c** Kaplan–Meier overall survival curves for three clusters of patients with distinct gene expression profiles. **d** Kaplan–Meier overall survival and event free survival curves for patients with high TERT and low TERT expression.

profiling ($n = 247$). Using the same gene list and clustering method as described for the RNA-seq dataset, we identified four clusters of tumors for the microarray dataset: clusters 1, 2 and 4 are ADRN-like, and cluster 3 is MES-like (Supplementary Fig. 9a, b). Again, two ADRN-like clusters, clusters 1 and 4, manifested the most benign and the poorest outcomes, respectively, while the MES-like cluster was intermediate in prognosis (Supplementary Fig. 9c). Similar to results from the RNA-seq dataset, low TERT expression in the microarray dataset is associated with better EFS and OS, while high TERT expression is associated with poorer EFS and OS (Supplementary Fig. 9d). Therefore, both TMM and tumor cell lineage may contribute to NB malignancy and could help refine prognosis.

## Discussion

The key discovery of the current study is the strong mechanistic connection between telomere regulation and tumor cell differentiation. The potential basis for this connection and its implications for the natural history of NB are discussed below.

One important outcome of this study is the discovery of a tight mechanistic relationship between telomerase activity and cell lineage in MYCN-amplified NB tumors. This relationship holds for both naturally-derived and experimentally-generated ADRN/

MES cell line pairs, with high telomerase indicative of the ADRN, and low telomerase indicative of the MES differentiation states, respectively. The Dn-hTERT experiment, in particular, supports a *causal* role of telomerase activity in regulating NB differentiation. Interestingly, this does not seem to apply to non-*MYCN*-amplified tumors, as evidenced by similar telomerase levels in the SH-SY5Y and SH-EP1 cells. The special importance of telomerase in *MYCN*-amplified tumors may be due to reciprocal effects of these proteins on each other's expression. In *MYCN*-amplified NB, MYCN is known to positively regulate TERT transcription[14], and in this study we demonstrated a reciprocal, positive effect of telomerase activity on MYCN expression. Since MYCN has previously been implicated in NB differentiation[6], the positive feedback loop between MYCN and telomerase in this subset of high-risk tumors may explain the dependence of NB differentiation on telomerase. Conversely, we speculate that in non-*MYCN*-amplified tumors, MYCN and telomerase may be excluded from the gene expression network that regulates ADRN/MES switch. These considerations suggest that the specific genetic alterations that trigger NB tumorigenesis have an impact on differentiation program in the tumor.

Our discovery of the positive feedback relationship between MYCN and TERT reinforces the concept of "feed forward loop" in driving the reprogramming of NB tumor cell lineages[9].

Notably, inducible expression of multiple transcription factors, including NOTCH paralogs and PRRX1, have been shown to drive the conversion of ADRN into MES cells, arguing against the idea of a single master regulator[9,10]. Instead, factors activated during the conversion process may be mutually reinforcing, either through positive feedback of transcription (as we have demonstrated for MYCN and telomerase) or through alternative mechanisms such as mutual protein stabilization (e.g., see[46]). As pointed out earlier, this type of feed forward network has the advantage of enabling rapid transitions between two meta-stable states of differentiation[9].

While the specific signals and pathways by which telomerase activity control MYCN expression and NB differentiation remain to be determined, the length of telomeres is evidently not an important factor. For example, in BrdU-treated BE(2)N and BE(2)C cells, the gradual conversion to MES cells paralleled the progressive loss in telomerase activity, yet no alterations in telomere lengths were observed. Likewise, the reversion to ADRN cells in late passages of Dn-hTERT-treated cells is accompanied by derepression of telomerase activity but not any detectable change in telomere lengths. Besides establishing the critical role of telomerase in differentiation, these results underscore the potential contributions of additional factors to telomere length homeostasis. Notably, both TRF1 and TRF2 are important regulators of telomere lengths, and both were downregulated during BrdU and Dn-hTERT-induced cell lineage conversion, thereby complicating the relationship between telomerase activity and telomere lengths. TRF1 is part of the "protein-counting", negative feedback loop that regulates telomerase access to telomeres[47,48]. As such, low TRF1 level in MES cells may enable greater access of telomerase, thereby compensating for the reduction in telomerase level. TRF2, on the other hand, has been shown to regulate the extent of telomere loss; high TRF2 level is associated with faster and occasionally drastic telomere shortening[49,50]. Therefore, the low TRF2 level in MES could compensate for low telomerase activity by mitigating telomere loss. It will be interesting in the future to investigate whether these alterations in telomere protein levels are part of the normal mechanism that neural lineage cells utilize to maintain telomere lengths during neural development, when telomerase activity is known to undergo drastic changes[35,51].

Consistent with our results, the inhibition of telomerase activity by Dn-hTERT in another MYCN-amplified cell line IGR-N-91 was reported to convert cells from a neuron-like morphology to one that resembles MES cells[52]. However, the repression of telomerase in this earlier study was stable, and no phenotypic reversion was observed. Because detailed transcriptional profiling was not available, it was unclear how similar the Dn-hTERT-induced cells were to the bone fide MES cell lineage. Nevertheless, when consider together with the current report, the data suggest that the tight causal relationship between telomerase and cell lineage may be applicable to MYCN-amplified NB tumors in general.

Telomerase is unlikely to be the only telomere factor involved in NB tumor cell differentiation, given the differences between ADRN and MES cells in the levels of many other telomere proteins. These include key components of the shelterin complex (TRF1, TRF2, TPP1, and POT1), as well as multiple factors involved in telomere maintenance (RTEL1 and ATRX). Indeed, previous studies have already implicated the double-strand telomere binding protein TRF2 in neural development, providing a plausible mechanism for how it could affect NB tumor cell differentiation. More specifically, TRF2 was shown to promote neural progenitor cell development and differentiation into mature neuron by interacting with REST (RE-1 silencing transcription factor) and antagonizing its neural repressive

function[31–33]. In support of this nontelomeric mechanism, we detected mostly cytoplasmic localization of TRF2 in ADRN cells. Interestingly, inspection of RNA-seq data revealed upregulation of REST mRNA in both BrdU- and Dn-hTERT-induced MES cells, supporting the potential involvement of the TRF2-REST pathway in NB differentiation. Therefore, like telomerase, TRF2 and other telomere proteins may also be part of the feed forward network that pormotes NB lineage conversion. Yet another differentially expressed telomere protein that can potentially affect disease biology is ATRX, a chromatin remodeling factor frequently inactivated in ALT-positive NB. Loss of ATRX is believed to trigger progressive telomere replication dysfunction, culminating ultimately in the activation of ALT during immortalization[53]. We found that ATRX is dramatically downregulated in MES cells, suggesting that these cells may be prone to telomere dysfunction and to the activation of ALT. Beyond NB, ALT is known to occur at higher frequencies in cancers of mesenchymal origin[23]. Whether this is linked to downregulation of ATRX is an interesting question for future investigation.

Previous analyses of ADRN and MES cells have not revealed strong differences in the levels of telomere-related factors. Our preliminary data suggests that some of the differences in protein levels may be due to post transcriptional regulation, and therefore overlooked in transcriptomic analysis. For example, RNA-seq data indicate that the levels of TRF1 and ATRX mRNA in BE(2)N was unchanged following BrdU treatment, in clear contrast to the drastic reduction in protein levels. Previous studies of neuroprogenitor cells also suggest post-transcriptional regulation of telomere-related genes during neural differentiation[33]. In addition, the discordance between ATRX mRNA and protein levels is supported by recent profiling of ALT-positive NB tumors[54]. More detailed proteomic analysis of different cell lineages may be required to gain a full understanding of the phenotypic differences between these lineages.

Both telomere maintenance and tumor cell lineage have been implicated in the prognosis of NB, each with plausible underlying mechanisms. Telomere DNA is subject to continuous erosion and drastic truncation owing to incomplete replication and telomere trimming. Without compensatory mechanisms, the loss of telomere DNA should eventually trigger a DNA damage response that compromises tumor cell proliferation. Not surprisingly, in two large-scale studies of NB tumor samples, the presence of either high telomerase activity or ALT activity is strongly associated with poor prognosis[18,19]. The role of NB tumor cell differentiation in disease progression is also plausible but mechanistically complex. Initial studies suggest that MES cells may be less tumorigenic in mice[7], whereas a more recent study reported comparable malignancy of matched ADRN/MES cell lines generated by controlled expression of NOTCH3-IC[9]. Importantly, these studies did not take into account the potential effects of host immune response, and growing evidence suggests that ADRN and MES cells differ dramatically in their ability to induce immunity and inflammation. Wolpaw et al. report elevated TLR signaling in natural and induced MES cells; upon stimulation, these cells secrete high levels of pro-inflammatory cytokines, leading to increased tumor cell killing by T-cells in vitro[37]. Similarly, Sengupta et al. demonstrated enrichment of innate and adaptive immune gene signature in MES cells, and showed that these cells not only engaged cytotoxic and natural killer cells, but also induced immune cell infiltration in immunocompetent mice[36]. Echoing these results, we show in our analysis strong upregulation of several key components of the DNA sensing/innate immunity pathway in naturally derived as well as BrdU- and Dn-hTERT-induced MES cells. In addition, the high levels of cGAS, STING and IRF3 in MES cells evidently enable these cells to generate a stronger response to viral DNA

challenge. Together, these studies highlight a strong linkage between telomere-remodeling, cell lineage reprogramming, and alterations in immunogenicity, which have strong implications for developing new biomarkers and therapies for this aggressive pediatric cancer. Indeed, by using a gene signature that incorporates differentially expressed genes related to telomere regulation, cell lineage, and immunogenicity, we were able to identify three clusters of NB tumors that manifest either a predominantly MES-like or ADRN-like phenotype. The different survival statistics for patients in different clusters illustrates the potential prognostic value of genes associated with these pathways. Most importantly, the strong interconnections between telomeres, cell lineage and immunity suggest that there may be many targets in the differentiation network of NB that can be manipulated to improve therapeutic response. In this regard, it is possible that inhibiting telomerase in *MYCN*-amplified tumors may prove especially efficacious—by simultaneously eliminating telomere maintenance and enhancing immunogenicity through tumor cell lineage reprogramming. However, the apparently greater chemoresistance of MES cells must also be taken into account in evaluating this and other related approaches.

## Methods
**Cell culture**. Neuroblastoma cell lines SK-N-BE(2)C (abbreviated as BE(2)C), and SK-N-HM were originally established at MSKCC, while SK-N-BE(2)N (abbreviated as BE(2)N), LA1-55N, LA1-5S, LA1-66N, LA1-6S, SH-SY5Y, and SH-EP1 cell lines were obtained from Robert Ross, Fordham University, Bronx, NY. These cell lines were cultured in RPMI-1640 medium (CORING) supplemented with 10% fetal bovine serum (FBS) (GEMINI) and 1% penicillin/streptomycin (Pen/Strep) (Gibco). The human cervical cancer cell line HeLa wase also obtained from ATCC, and was cultured in Dulbecco's modified Eagle's medium (DMEM) (CORING) supplemented with 10% FBS and 1% Pen/Strep. Phoenix cell line for retrovirus packaging was kindly provided by Dr. Xin-Yun Huang at Weill Cornell Medicine, and was cultured in DMEM supplemented with 10% FBS, 1% Pen/Strep, and 1% glutamine.

**Production of retrovirus stocks and infections of NB cell lines**. pBABE-puro-hTERT and pBABE-puro-Dn-hTERT plasmid (dominant negative hTERT) was provided by Dr. Duncan Baird (Cardiff University, U.K.)[44]. Amphotropic retroviruses were generated in Phoenix packaging cells that express gag-pol and envelop proteins by $Ca_2PO_4$ transfection protocols, which is slightly modified from the protocols described by Dr. Garry Nolan (Stanford University) and Dr. Titia de Lange (Rockefeller University). Briefly, 20 μg of each retroviral vector (pBABE-puro-hTERT and pBABE-puro-Dn-hTERT) was mixed with 250 ul of 1 M $CaCl_2$ in total volume of 500 μl. 500 μl of 2x HBS buffer (50 mM of HEPES (pH 7.05), 10 mM of KCl, 12 mM of Dextrose, 280 mM of NaCl, 1.5 mM of $Na_2HPO_4$ was added to the mixture in order to produce working $CaPO_4$ precipitates. HBS/DNA solution was added dropwise to the cells ($5 \times 10^6$) in the medium in a 10 cm plate and the plate was swirled gently to evenly distribute DNA/$CaPO_4$ particles. At 20 hr post-transfection, the medium was replaced with fresh medium. Transfection efficiency was monitored based on GFP expression. At 48 hr post-transfection, the viral supernatants were filtered through a 0.45 μm filter and used to infect BE(2)N and BE(2)C cells with the addition of 4 μg/ml polybrene. Cells were infected twice for 6 ~ 8 h, and were selected with puromycin (0.9 μg/ml for BE(2)N and 0.75 μg/ml for BE(2)C)) at 48 hr post-infection.

**Western**. PVDF membrane filters with transferred proteins were blocked with TTBS containing 5% non-fat milk, and incubated with the following primary antibodies: mouse anti-TRF1 (SCBT, sc-56807, 1:3000), mouse anti-TRF2 (Novusbio, NB100-56506, 1:3000), mouse anti-RAP1 (SCBT, sc-53434, 1:3000), mouse anti-TIN2 (Novusbio, NB600-1522, 1:2000), rabbit anti-TPP1 (Bethyl, A303-069A-T, 1:2000), rabbit anti-POT1 (Novusbio, NB500-176, 1:3000), mouse anti-GAPDH (ABclonal, AC002, 1:100,000), mouse anti-TERT (SCBT, sc-393013, 1:1000), mouse anti-STN1/OBFC1 (SCBT, sc-374178, 1:2000), mouse anti-PRIM1 (SCBT, sc-390265, 1:3000), mouse anti-POLA2 (SCBT, sc-398255, 1:3000), mouse anti-RPA70 (SCBT, sc-48425, 1:1500), rabbit anti-TZAP (Proteintech, 24665-1-AP, 1:2000), mouse anti-ATRX (SCBT, sc-55584, 1:3000), mouse anti-RTEL1 (SCBT, sc-515427, 1:3000), mouse anti-RIF1 (SCBT, sc-515573, 1:3000), mouse anti-MYCN (SCBT, sc-53993, 1:3000), mouse anti-IRF3 (SCBT, sc-33641, 1:3000), rabbit anti-STING (Cell Signaling, #13647, 1:3000), mouse anti-cGAS (SCBT, sc-515777, 1:3000), mouse anti-SLUG (SCBT, sc-166476, 1:2000), mouse anti-GATA3 (SCBT, sc-269, 1:2000), rabbit anti-phospho-STAT1 (Cell Signaling, #9167, 1:3000), rabbit anti-phospho-TBK1 (Cell Signaling, #5384, 1:3000). Secondary antibodies were either HRP-linked anti-rabbit IgG (Cell Signaling, #7074, 1:5000) or HRP-linked Anti-Mouse IgG (Cell Signaling, #7076, 1:5000), and were applied

to the filters in TTBS containing 5% non-fat milk. After washing, the blots were visualized using chemiluminescence reagents (Tanon™ High-sig ECL Western Blotting Substrate, ABclonal). The antibody information is also summarized in Supplementary Table 2.

**IF-FISH**. For Immunofluorescence (IF) combined with fluorescence in situ hybridization (FISH)[55], LA1-55N, LA1-5S, and HeLa cells were seeded at $7 \times 10^5$, $4 \times 10^5$, or $3 \times 10^5$ cells per well, respectively, in 6-well plates containing coverslips with Poly-D-Lysine (PDL) coating (Neuvitro Corp.), and cells were cultured for 24 h. Cells were fixed with 4% paraformaldehyde in PBS for 10 min, permeabilized with ice-cold 0.5% Triton X-100 on ice for 5 min, and stained for protein antigen overnight at 4 °C in a humidified chamber. The following primary antibody were used: mouse-α-TRF1 (TRF-78, ab10579, Abcam; 1:100), mouse-α-TRF2 (NB100-56506, Novus Biologicals; 1:200). Goat-α-mouse-Alexa-488 (A28175, Invitrogen; 1:500) was used as the secondary antibody. Cells were fixed again with 4% paraformaldehyde in PBS for 6 min, and then subjected to FISH using a peptide nucleic acid (PNA) probe specific to the telomeric sequence $(CCCTAA)_3$ (TelC-Cy3, PNA BIO). Cells were dehydrated successively in 70%, 90%, and 100% ethanol and air dried, and nuclear DNA was denatured for 10 min at 85 °C in hybridization buffer containing 1.25 μg/ml $(C_3TA_2)_3$-Cy3-labeled PNA telomeric probe, 70% formamide, 0.5% blocking reagent (Roche), 10 mM Tris-HCl (pH 7.4). After denaturation, incubation was continued overnight at 4 °C in a humidified chamber. Cells were washed twice, each for 15 min with 70% formamide/10 mM Tris-HCl (pH 7.4), followed by a 5 min wash with PBS three times. After air-drying, the coverslips were placed on top of embedding medium (ProLong Gold antifade reagent with DAPI, Invitrogen) on a slide and sealed with nail polish. FLUOVIEW FV10i (OLYMPUS) microscope was used to capture images. For BrdU-treated BE(2)N, cells were cultured with 7.5 μM of BrdU for up to 16 days. On the day before the IF-FISH procedure, cells were seeded at $7.5 \times 10^5$ cells per well in 6-well plates containing coverslips with PDL coating. The cells were allowed to grow on the coverslips for another 24 hr, and then subjected to IF-FISH.

**Immune activation**. KOS and vH609 virus particles were prepared from the medium of Vero cells[45]. Briefly, $1.5 \times 10^8$ Vero cells were infected overnight (18 h at 37 °C) at an MOI of 5 PFU per cell. Infected cells were scraped into the cell medium, and 5 M NaCl was added to a final concentration of 0.5 M NaCl. Cells were pelleted, and the medium was transferred to SW28 rotor tubes; virions were pelleted out of the medium by centrifugation at 20,000 rpm for 35 min. The resulting pellet was resuspended in 100 μl of phosphate-buffered saline (PBS) plus protease inhibitors. The samples were treated with DNase I at room temperature for 30 min, layered on top of a 20 to 50% sucrose gradient in TNE buffer (10 mM Tris, 150 mM NaCl, and 1 mM EDTA, pH 7.5), and then spun in SW41 rotor at 24,000 rpm for 1 h. The virion band was collected, transferred to another SW41 tube, and diluted 1:5 with TNE buffer, followed by centrifugation to collect the virions. The virions were resuspended in TNE buffer, and the titer of each preparation was determined. To test immune response to viruses, neuroblastoma cells were cultured in a 12-well dish to a density of ~$5 \times 10^5$ cells per well, and then treated with KOS or vFH609 virus particles at a titer of ~$3 \times 10^3$ virus particles per cell. Cells were collected 4 h after infection, and extracts were subjected to Western analysis.

**RNA-seq**. RNA-sequencing was performed on the following human NB cell lines: LA1-66N, LA1-6S, BE(2)N (untreated), BE(2)N (Dn-hTERT treated—on day 33, day 55, and day 82), BE(2)N (DMSO-treated), and BE(2)N (BrdU treated—on day 5, day 11, day 20, and day 26). Total RNA was isolated with the RNeasy Mini kit (Qiagen), and the quality of the preparations assessed on an Agilent 2100 Bioanalyzer (Agilent Technologies). TruSeq stranded mRNA libraries were prepared and sequenced with paired-end 50 bps on Illumina NovaSeq 6000 by the WCM Genomics Core Facility. The raw sequencing reads in BCL format were processed through bcl2fastq 2.19 (Illumina) for FASTQ conversion and demultiplexing. RNA reads were aligned and mapped to theGRCh37 human reference genome by STAR (Version2.5.2) (https://github.com/alexdobin/STAR)[56], and transcriptome reconstruction was performed by Cufflinks (Version 2.1.1) (http://cole-trapnell-lab.github.io/cufflinks/). The abundance of transcripts was measured with Cufflinks in Fragments Per Kilobase of exon model per Million mapped reads (FPKM)[57,58]. The fold changes were calculated by directly comparing the FPKM values of samples and controls.

Up- and downregulated genes were clustered and displayed as heatmaps using ClustVis[59]. For identification of overlaps between gene sets, the lists were uploaded and analyzed at the Bioinformatics Evolutionary Genomics server (http://bioinformatics.psb.ugent.be/webtools/Venn/). Statistical significance values were calculated using the Nematode Bioinformatics server (http://nemates.org/MA/progs/overlap_stats.html). The data series have been uploaded to GEO (accession GSE171404).

**RT-PCR**. Total RNA was isolated from Dn-hTERT expressing BE(2)N cells collected at days 33, 55, and 82 (D33, D55, D82), using the RNeasy Mini kit (Qiagen). Purified RNA was reverse transcribed to cDNA using Transcriptor Reverse Transcriptase (Roche), and the cDNA products subjected to PCR using 1 μl cDNA,

2.5 U Choice-Taq Blue DNA polymerase (Denville) and PCR primers (2.5 μM) in a total volume of 20 μl. Each cDNA obtained from Dn-TERT expressing cells were subjected to serial dilutions to facilitate quantitative comparison. Relative expressions of mRNAs (for both wild type and Dn-hTERT) were calculated by comparing the intensities and controlling for dilution factors using GraphPad Prism.

**Assays of telomeres and telomerase activity (TRF Southern, STELA, in-gel hybridization, TRAP).** For telomere restriction fragment (TRF) Southern analysis, chromosomal DNAs were digested with *Hinf*I and *Rsa*I, and fractionated in 0.8% agarose gels. Following transfer to nylon membranes, the blot was hybridized to a telomere repeat probe ((TTAGGG)$_{82}$)[60]. To determine average telomere lengths, we used the weighed average method[61,62]. For STELA analysis, chromosomal DNAs were ligated to telorette oligos[63] in 15 μl reaction containing 10 ng genomic DNA, 0.01 μM telorette oligo, 1x CutSmart Buffer (NEB), 1 mM ATP, and 800 U T4 DNA ligase (NEB) at 35 °C for 20 h[64]. PCR assays were carried out in 15 μl reactions containing 500 pg template DNA, 1x Taq buffer (Thermo Fisher), 0.3 mM of each dNTP, 0.4 μM of forward and reverse primers, and 1 U of Pwo:Taq (10:1) polymerase mix. Twenty-nine cycles of 15 sec at 94 °C, 30 sec at 64 °C and 10 min at 68 °C were performed. To ensure adequate coverage of the telomere size distribution, we performed 3−6 parallel PCR reactions for each ligated DNA sample. The reaction products were analyzed by electrophoresis in 0.8% agarose gels and subjected to Southern using the appropriate subtelomeric probes or a telomere repeat probe ((TTAGGG)$_{82}$). Following PhosphorImager scanning (GE Healthcare), the sizes of individual telomeres were determined using TESLA software[64], and the results analyzed and plotted using Prism (GraphPad Software). For in-gel hybridization analysis, untreated genomic DNA was loaded directly onto a 0.8% agarose gel, and the current applied until the bromophenol blue dye has migrated about 4 cm into the gel[65]. The gel was soaked in 2x SSC, and then dried for 30 min (into a thin layer) using a vacuum pressure of 550 mm Hg. The labeled G4 and C4 oligonucleotides, corresponding to four copies of the telomeric G-strand and C-strand repeats, were used as the probes, and hybridization was performed in the Church Mix at 42 °C.

For telomere repeat amplification protocol (TRAP) assays, the reaction mix contained 20 mM Tris.HCl, pH 8.3, 1.5 mM MgCl$_2$, 63 mM KCl, 0.05 % Tween-20, 1 mM EGTA, 50 μM each of dATP, dCTP, dGTP, TTP, 2 ng/μl TS primer (5′ labeled with gamma-P$^{32}$-ATP), 2 ng/μl NT primer, 2 ng/μl ACX primer, 0.0002 pM TSNT oligonucleotide, 2U choice-Taq DNA polymerase, and varying amounts of extract[66]. The telomerase reaction was allowed to proceed at 25 °C for 45 min followed by 28 cycles of thermocycling (30 s at 95 °C, 30 s at 52 °C and 45 s at 72 °C). The reaction products were analyzed by electrophoresis in 10% polyacrylamide gel (1x TBE) and PhosphorImager scanning. To determine relative telomerase levels between samples, we first calculated the ratio of TRAP signals to the control PCR product (IC) for each assay. A standard curve that relates extract amounts to the TRAP/IC ratios was then constructed using the dilution series that yielded the highest signals in a given gel. (Typically the curve is close to linear or hyperbolic.) A series of XY values were then generated from the curve using PRISM, which allowed us to determine the equivalent amount of extract needed to yield the specific TRAP/IC signals produced by other samples in the gel. Multiplying this value by the dilution factor provides the relative telomerase activity level in a given extract.

**Oligonucleotides.** The oligonucleotides used in the current study are as follows: TTAGGG$_4$ (G4), TTAGGG TTAGGG TTAGGG TTAGGG; CCCTAA$_4$ (C4), CCCTAA CCCTAA CCCTAA CCCTAA; TTAGGG$_8$ (G8), TTAGGG TTAGGG TTAGGG TTAGGG TTAGGG TTAGGG TTAGGG TTAGGG; CCCTAA$_8$ (C8), CCCTAA CCCTAA CCCTAA CCCTAA CCCTAA CCCTAA CCCTAA CCCT AA; TS, AAT CCG TCG AGC AGA GTT; ACX, GCG CGG CTT ACC CTT ACC CTT ACC CTA ACC; NT, ATC GCT TCT CGG CCT TTT; TSNT, AAT CCG TCG AGC AGA GTT AAA AGG CCG AGA AGC GAT; C Telorette 1, GCTC CGTGCATCTGGCATCCCCTAAC; C Telorette 2, GCTCCGTGCATCTGGCAT CTAACCCT; C Telorette 3, GCTCCGTGCATCTGGCATCCCTAACC; C Telorette 4, GCTCCGTGCATCTGGCATCCTAACCC; C Telorette 5, GCTCCGTGC ATCTGGCATCAACCCTA; C Telorette 6, GCTCCGTGCATCTGGCATCACC CTAA; Teltail, GCTCCGTGCATCTGGCATC; XpYp1, GTTGTCTCAGGGT CCTAGTG; XpYpB2, TCTGAAAGTGGACC(A/T)ATCAG; DnTERT-RTPCR-F1, G TAC TTT GTC AAG GTC GCG A; TERT-RTPCR-F1, G TAC TTT GTC AAG GTG GAT G; TERT-2300-RGTC AAG GTA GAG ACG TGG CT. This information is also summarized in Supplementary Table 2.

**NB tumor database analysis.** The Pediatric Neuroblastoma (Target, 2018) tumor collection at cBioportal were profiled with respect to the mRNA levels of 29 genes implicated in telomere, cell lineage, and immune regulation and that showed strong differential expression in ADRN and MES cells (Fig. 6b). Tumors with RNA-seq ($n = 137$) and microarray ($n = 247$) expression data were analyzed separately. Gene expression levels were rank transformed and clustered using Partition around medoids. The optimal numbers of clusters were determined based on the average silhouette width and strength of association with patient overall survival (OS) and were found to be 3 and 4 for the RNA-seq and microarray datasets, respectively. The relationships between the expression clusters and patients' overall and event free survival (EFS) were evaluated using log-rank test. To identify genes whose expression showed consistent differences between cluster 1 and 3 patients, we applied the nonparametric Wilcoxon rank sum test to each of the 29 genes in the signature and used $P < 0.01$ for both tests as the cutoff (Supplementary Table 1). To assess the association between TERT levels and survival, TERT RNA levels were converted to a categorical variable (high TERT and low TERT) dichotomized at the median. Overall survival and event free survival were then examined in (i) univariate analysis using log-rank test and (ii) multivariable analysis adjusting for age and stage using Cox proportional hazards model. These analyses were carried out using R 4.0.4 (https://r-project.org).

**Statistics and reproducibility.** Statistical calculations including two-tailed Student's tests were performed using GraphPad Prism 9. Where they are relevant to the conclusions, the p-values were presented in numerical forms. Each experiment was repeated two or three times to ensure reproducibility. The differences between BrdU and Dn-hTERT treated samples were confirmed using two or three independently generated series of samples.

**Reporting summary.** Further information on research design is available in the Nature Research Reporting Summary linked to this article.

## Data availability

The datasets generated and/or analyzed during the current study are included within the paper and Supplementary Information or available from the corresponding author upon reasonable request. The original, uncropped gel and blot images for Figs. 1–6 are included in the Supplementary Information file as Supplementary Fig. 10. The source data for all the plots in the paper are included in the Supplementary Data 1 file. The RNA-seq data associated with Fig. 3, Fig. 4 and Supplementary Fig. 7 have been uploaded to GEO (accession GSE171404).

## Code availability

All the software used in this study is publicly available.

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

## Acknowledgements

We thank the Computational Genomics Core at WCM for the RNA-seq analyses, and members of our laboratories for comments. This work was supported by a Seed Grant for Collaborative Multi-Investigator Projects between Cornell University Ithaca, Weill Cornell Medicine, and Cornell Tech. N.F.L is a recipient of the William Randolph Hearst Endowed Faculty Fellowship in Microbiology. X.K.Z. is supported in part by 1UL1 TR002384-01 to Weill Cornell Clinical and Translational Science Center.

## Author contributions

E.Y.Y., X.K.Z., N.-K.C., and N.F.L. conceived the study. E.Y.Y., N.F.L., E.F.-P., and X.K.Z. designed the experiments and computational analysis. E.Y.Y., S.S.Z., S.A., E.F.-P., X.K.Z., and N.F.L performed the experiments and analyzed the data. E.Y.Y. and N.F.L. wrote the paper with advice from X.K.Z. and N.-K.C.

## Competing interests

The authors declare the following competing interests: N.K.C. and MSKCC have financial interest in Y-mAbs, Abpro-Labs and Eureka Therapeutics. N.K.C. reports receiving commercial research grants from Y-mabs Therapeutics and Abpro-Labs Inc. N.K.C. was named as inventor on multiple patents filed by MSKCC, including those licensed to Ymabs Therapeutics, Biotec Pharmacon, and Abpro-labs. N.K.C. is a SAB member for Abpro-Labs and Eureka Therapeutics. Link to COI at MSKCC: https://tinyurl.com/y3vn7opn. All other authors declare no competing interests.
