## [Transparent Peer Review File · Communications Biology]

Reviewers' comments:

Reviewer #1 (Remarks to the Author):

The manuscript reciprocal impacts of telomerase activity and tumour cell differentiation in neuroblastoma tumor biology by Young Yu et al is well written and establishes a link between telomere maintenance and differentiation – both important areas of neuroblastoma biology.

Specific comments;

Page 13: In paragraph 2 the authors state there is “better OS for patients with low levels of TERT”. Which is concordant with figure 6d. However in paragraph 3 it says “low TERT expression is associated with a poorer EFS and OS” – I assume this is a typo as it is not concordant with the data in supp figure 8 where low TERT is associated with better survival?

Page 18, Final paragraph of discussion: The authors conclude that inhibiting telomerase in MYCN-amplified tumours may be an especially potent strategy. The data presented in the paper does not necessarily support this – the authors show that this may enhance immunogenicity which indeed may represent an important therapeutic approach, however they also show that down regulation of telomerase is associated with a switch to a mesenchymal state, known to be associated with therapy resistance. I would suggest either deleting the last sentence or amending it accordingly.

Figure 1 uses black text to denote ADRN cell lines and green to denote MES cell lines, this should be explicitly stated in the figure legend to make it easier to understand and the same colours should be used throughout (e.g. figure 2 uses blue and green text)

Supp figure 3a: there needs to be a loading control shown here (e.g. GAPDH) for the western blot

Reviewer #2 (Remarks to the Author):

Yu and colleagues describe a relation between telomerase activity and MES/ADRN lineage differentiation in the childhood tumor neuroblastoma. This work merges two important research fields in neuroblastoma, i.e. telomere regulation (Peifer et al., *Nature*, 2015; Valentijn et al., *Nat. Genet.*, 2015) and tumor cell lineages (Boeva et al., *Nat. Genet.*, 2017; van Groningen et al., *Nat. Genet.*, 2017). The authors extensively characterize telomerase pathways and telomeres in pairs of naturally occurring MES and ADRN cell types and in induced MES cells. In general, the paper is well written.

Major Comment:

The authors do not identify major changes in telomere length between MES and ADRN cells. However, this may be due to the fact that MES and ADRN cell lineages can switch between MYCN/TERT/telomerase dependency in ADRN cells and the ALT pathway in MES cells. Did the authors investigate this, and could they provide experimental analysis for this hypothesis?

Can the authors comment further on the regulation of telomere length during ADRN to MES transition? Is telomere length transiently downregulated during the reprogramming process?

In the BrdU-reprogramming experiment, the authors claim a transition from ADRN to MES cells. However, since neuroblastoma cell lines can be composed of both MES and ADRN cell types, an alternative explanation is that MES cells have been selected, rather than induced. Can the authors provide evidence that MES cells are induced instead of selected?

Reviewer #1 (Remarks to the Author):

The manuscript reciprocal impacts of telomerase activity and tumour cell differentiation in neuroblastoma tumor biology by Young Yu et al is well written and establishes a link between telomere maintenance and differentiation – both important areas of neuroblastoma biology.

Specific comments

Page 13: In paragraph 2 the authors state there is “better OS for patients with low levels of TERT”. Which is concordant with figure 6d. However in paragraph 3 it says “low TERT expression is associated with a poorer EFS and OS” – I assume this is a typo as it is not concordant with the data in supp figure 8 where low TERT is associated with better survival?

This was an error and has been corrected.

Page 18, Final paragraph of discussion: The authors conclude that inhibiting telomerase in MYCN-amplified tumours may be an especially potent strategy. The data presented in the paper does not necessarily support this – the authors show that this may enhance immunogenicity which indeed may represent an important therapeutic approach, however they also show that down regulation of telomerase is associated with a switch to a mesenchymal state, known to be associated with therapy resistance. I would suggest either deleting the last sentence or amending it accordingly.

The last two sentences of the paragraph have been amended as suggested by the referee.

Figure 1 uses black text to denote ADRN cell lines and green to denote MES cell lines, this should be explicitly stated in the figure legend to make it easier to understand and the same colours should be used throughout (e.g. figure 2 uses blue and green text)

The color code (blue for ADRN and green for MES cells) is now explained in Figure 1 legend and used consistently in throughout.

Supp figure 3a: there needs to be a loading control shown here (e.g. GAPDH) for the western blot

The loading controls have been added (Supp. Fig. 4a).

Reviewer #2 (Remarks to the Author):

Yu and colleagues describe a relation between telomerase activity and MES/ADRN lineage differentiation in the childhood tumor neuroblastoma. This work merges two important research fields in neuroblastoma, i.e. telomere regulation (Peifer et al., Nature, 2015; Valentijn et al., Nat. Genet., 2015) and tumor cell lineages (Boeva et al., Nat. Genet., 2017; van Groningen et al., Nat. Genet., 2017). The authors extensively characterize telomerase pathways and telomeres in pairs of naturally occurring MES and ADRN cell types and in induced MES cells. In general, the paper is well written.

Major Comment

The authors do not identify major changes in telomere length between MES and ADRN cells. However, this may be due to the fact that MES and ADRN cell lineages can switch between MYCN/TERT/telomerase

dependency in ADRN cells and the ALT pathway in MES cells. Did the authors investigate this, and could they provide experimental analysis for this hypothesis?

We tested the levels of c-circles (a marker of ALT activity) in BrdU-treated BE(2)C cells and did not observe significant changes, arguing against the idea that telomere maintenance was due to the activation of ALT (Supp. Fig. 4d).

Can the authors comment further comment on the regulation of telomere length during ADRN to MES transition? Is telomere length transiently downregulated during the reprogramming process?

As far as we can measure, there was no decrease of telomere lengths at any time point tested during the BrdU-mediated conversion (Fig. 3d and Supp. Fig. 4c).

In the BrdU-reprogramming experiment, the authors claim a transition from ADRN to MES cells. However, since neuroblastoma cell lines can be composed of both MES and ADRN cell types, an alternative explanation is that MES cells have been selected, rather than induced. Can the authors provide evidence that MES cells are induced instead of selected?

We have monitored simultaneously cell proliferation and morphologic conversion during the BrdU-induced re-programing process. Our data indicate that while proliferation largely ceased after 8 days, there was still a progressive change in the morphology of cells toward the MES phenotype (flattened and enlarged), supporting the idea that the MES cells are induced rather than selected (Supp. Fig. 2a and 2b).

Reviewers' comments:

Reviewer #1 (Remarks to the Author):

all comments addressed

Reviewer #2 (Remarks to the Author):

The authors added some experimental work to my previous comments. Unfortunately however, these analyses are not convincingly answering my previous comments.

Major concerns:

Previous comment 1:

Although the authors added data of c-circles as a marker of ALT activity in BrdU-treated SKNBE2c cells, this does not allow the general conclusion of a lack of ALT activity in stable, unselected MES cells. The authors have multiple isogenic pairs of MES and ADRN cells, which should be used for a direct comparison of ALT activity between the divergent MES and ADRN states in at least two cell line pairs.

Previous comment 3:

In my previous comment 3, I raised the concern that MES cells could have been selected, rather than induced by BrdU-treatment. The authors responded by monitoring simultaneously cell proliferation and morphologic conversion during this re-programing process.

Unfortunately, this experiment is not sufficient to support the claim that MES cells are induced from ADRN cells by treatment with BrdU. The authors should use specific markers to select ADRN cells and treat this pure ADRN population with BrdU after which they can test the induction of the MES state.

Reviewer #1 (Remarks to the Author):

all comments addressed.

Reviewer #2 (Remarks to the Author):

The authors added some experimental work to my previous comments. Unfortunately however, these analyses are not convincingly answering my previous comments.

Major concerns:

Previous comment 1:

Although the authors added data of c-circles as a marker of ALT activity in BrdU-treated SKNBE2c cells, this does not allow the general conclusion of a lack of ALT activity in stable, unselected MES cells. The authors have multiple isogenic pairs of MES and ADRN cells, which should be used for a direct comparison of ALT activity between the divergent MES and ADRN states in at least two cell line pairs.

We have included C-circle data on three matched pairs of ADRN and MES cell lines (Supp. Fig. 4, right panel), which did not show significant ALT activity.

Previous comment 3:

In my previous comment 3, I raised the concern that MES cells could have been selected, rather than induced by BrdU-treatment. The authors responded by monitoring simultaneously cell proliferation and morphologic conversion during this re-programing process.

Unfortunately, this experiment is not sufficient to support the claim that MES cells are induced from ADRN cells by treatment with BrdU. The authors should use specific markers to select ADRN cells and treat this pure ADRN population with BrdU after which they can test the induction of the MES state.

We appreciate the reviewer's comment. In the revised manuscript, we provided additional, semi-quantitative Western analysis of STING, an MES marker, during BrdU treatment. We showed that the marker expression increased significantly during the late stage of phenotypic conversion, when there was no cell proliferation. This, coupled with gradual morphologic change during the same time period, suggests that selection alone cannot explain the effects of BrdU on the lineage switch of the cell population. We also qualified our conclusion to indicate that some selection may occur in the initial period of BrdU treatment, which is associated with some proliferation (~ 4-5 PD). This possibility does not alter the major conclusion of the manuscript on the relationship between neuroblastoma cell lineage and telomere regulation.

We would also like to point out that the cell lines were initially cloned and passaged with differential culturing technique to maintain the ADRN phenotype (Ciccarone et al., Cancer Research 49, 219-225). While we could try to further purify the cell population with an ADRN marker, it is not guaranteed to generate a pure ADRN population. So in testing the "selection" hypothesis, we still need to consider the degree of cell proliferation, as we did in the revised manuscript.

REVIEWERS' COMMENTS:

Reviewer #2 (Remarks to the Author):

My concerns have been addressed by the authors. I have no further comments.